# A Multidimensional Analysis of Factors Impacting Mobility of Open-Access Multilane Highways

Jamal Ahmed Khan [1,*], Muhammad Bilal Khurshid [2], Arshad Hussain [1] and Asif Azam [3]

1 NICE, SCEE, NUST H-12, Islamabad 25000, Pakistan
2 Frontier Works Organization (FWO), Rawalpindi 46000, Pakistan
3 National Highway Authority (NHA) Headquarters, Islamabad 25000, Pakistan
* Correspondence: jahmed.tnphd15nit@student.nust.edu.pk; Tel.: +92-33-3578-3564

**Abstract:** Open-access multilane highways have a significant share in the highway network of Pakistan and other developing countries in Asia. These highways have high access density, design inconsistency, and other operational characteristics that differentiate them from partially controlled-access multilane highways. This study identified significant factors affecting the mobility of open-access highways based on road users' perceptions as well as field observations. An interview-based questionnaire survey from local respondents and an in-service road survey formed the database for the present study. Questionnaire survey results showed that heavy traffic was the most critical mobility influencing factor on open-access multilane highways, followed by road width and condition, whereas the result of multilinear regression revealed that the most significant variable was access density, followed by the flow and pedestrian crossings. However, it was concluded that controlling access density, preventing pedestrian crossings, and improving pavement condition will improve the mobility of open-access multilane highways.

**Keywords:** multilane highways; performance measures; mobility influencing factors; road user survey; regression analysis

## 1. Introduction

In the past few decades, rapid growth in traffic volume has impacted the mobility of multilane highways across the globe. Multilane highways in Pakistan are generally classified as Motorways (Access-controlled), Expressways (Partially controlled access), and National Highways (Open access). In addition to motorways, open-access multilane highways have a significant share in the highway network of Pakistan and other developing countries in Asia. Open-access highways form the backbone of the highway system in Pakistan, providing inter-provincial linkages along major corridors and playing an important role in trade and economic development. These highways function as arterials and collectors in local conditions to provide accessibility and mobility. National highway N-5 is one example of such a highway that stretches from one end of the country to another, carrying over 65% of the traffic load in Pakistan [1]. Open-access multilane highways differ from access-controlled and partially controlled-access highways, especially in developing countries. Partially controlled-access highways have moderate access density, designated bus stops, and designated pedestrian crossings, and all types of vehicles are allowed on such highways. In contrast, open-access highways have high access density, design inconsistency, and other operational characteristics (frequent pedestrian crossings, undesignated bus stops, highly heterogeneous traffic, and undesignated U-turns) that result in low operating speed (40–60 km/h). Such characteristics differentiate open-access multilane highways from other multilane highways. Various studies have also highlighted these differences and deficiencies in existing methodologies to analyze open-access highways [2–4].

Several research studies have investigated the impact of design and operational characteristics, such as pavement condition, environmental factors, traffic heterogeneity, driver,

and vehicle characteristics, on the mobility of highways using field data. However, open-access multilane highways, particularly in developing countries, need further exploration. Especially, there is a lack of research on the mobility of open-access multilane highways based on users' perceptions. While planning any highway facility, the users' opinions should be the primary consideration as it increases the user acceptance of the facility. It is also important to know how closely the professional judgment of designers and consultants corresponds to road-user perceptions as it can influence the highway operational analysis methodology. The same concept was emphasized in various studies, including the widely accepted highway capacity manual 2010 [5–8]. In recent years, studies have been conducted to assess road users' perspectives on essential factors in determining the quality of their trips [7,9,10]. However, most of these studies focused on multilane highways that usually have some level of access control, consistency in design, and homogeneous traffic conditions. The road users' opinions on open access and heterogeneous traffic conditions needs to be explored. Similarly, no comprehensive research has been conducted to determine the combined association between geometric design features and traffic flow on highway mobility along open-access multilane highways based on both road users' opinions and in-service road data. Moreover, it is also important to investigate the issues faced by practitioners in designing and managing open-access multilane highways, especially in developing countries, such as the use of the most appropriate mobility performance measure and significant influencing factors related to the mobility of open-access multilane highways based on road users' perception and based on field data of such in-service highways (including statistical modeling and analysis of these influencing factors). Although there is a severe lack of comprehensive and collective research on the aforementioned aspects, such critical issues of open-access multilane highways and other important issues related to asset management of these highways, especially in developing countries, have been identified in some past studies in the literature, in which practitioners' opinions have been obtained using questionnaire surveys [11–14]. Therefore, this study aimed to answer three research questions related to open-access multilane highways: (1) What are the most suitable mobility performance measures for analysis and decision-making on open-access multilane highways; (2) what are the significant factors affecting the mobility of open-access multilane highways based on road users' opinions, and (3) what are the significant factors affecting the mobility of open-access multilane highways in real-time field conditions (basing on field data of in-service highways).

## 2. Synthesis of Past Literature

Highway mobility can incorporate several quantitative elements, such as riding comfort and the absence of speed changes. However, the most basic function is operating speed and travel time [15]. The mobility of highways in this study refers to the ability to move quickly, easily, and economically on a highway at nearly free-flow conditions. An important step for evaluating highway performance is the selection of appropriate performance measures. Al-Kaisy and Jafari [16] suggested that the top three criteria for suitable performance measures on two-lane highways are: sensitivity to traffic conditions, sensitivity to road conditions, and relevance to road user perception. A study for establishing the Indonesian highway capacity manual mentioned speed as the primary measure of highway performance as it is simple to comprehend for both road users and highway agencies [17]. However, a similar study conducted on Indian highways by Arun and Madhu [18] stated that agreeing upon one service measure is difficult due to several heterogenetic characteristics in traffic operations. Alternatively, a widely accepted concept of the level of service (LOS) has been used as a qualitative measure for almost four decades. However, according to Choocharukul and Sinha [6], the highway capacity manual's use of traffic density as a single measure of performance for LOS does not precisely reflect what road-users perceive about road conditions. A list of mobility performance measures identified in the literature with a specific focus on their applicability on open-access highways has been summarized in Table 1.

**Table 1.** Mobility Performance Measures Identified from Past Literature.

| S/No | Performance Measure | S/No | Performance Measure | S/No | Performance Measure |
|------|---------------------|------|---------------------|------|---------------------|
| 1 | Travel time | 9 | Queue | 17 | Travel-time index |
| 2 | Speed | 10 | Stops | 18 | Platoon characteristics (critical headway for platoons) |
| 3 | Traffic density | 11 | Density | 19 | Volume/capacity |
| 4 | Travel rate | 12 | Delay rate | 20 | Corridor mobility index |
| 5 | Delay ratio | 13 | Level of service (LOS) | 21 | Delay per person |
| 6 | Percentage of delayed trips | 14 | Travel speed as % of free-flow speed | | |
| 7 | Congestion index: (weighted average) | 15 | Congestion (as % loss in freedom of movement) | | |
| 8 | Combination of service Measures | 16 | Speed reduction index | | |

References: [18–22].

A detailed literature review was also carried out to identify important factors that could impact the mobility of open-access multilane highways. According to Semeida [23], lane width, heavy vehicle percentage, and accessibility significantly influence lane capacity at tangent sections on open-access multilane rural roads in Egypt. Moreover, traffic volume composition is also significantly associated with highway capacity on open-access highways, especially in developing countries such as India and Sri Lanka [3,24]. Researchers also studied the impact of pavement condition on free-flow speed but only for controlled-access and partially controlled-access highways [25,26]. Another study by Salini and George [2] concluded that the buses stopping at bus stops also obstruct traffic flow on open-access multilane highways in India. Similarly, Dhamaniya and Chandra [27] concluded in their study that crossing pedestrians significantly reduce the capacity of the highway on open-access multilane highways. During the literature review, it was observed that a limited number of studies attempted to obtain road users' perception, and significantly fewer studies were related to open-access multilane highway mobility and its influencing factors [8,28]. However, according to the highway capacity manual, it is important to obtain road users' perspectives to assess how well a transportation facility functions [5].

Although an extensive review of more than 100 studies was conducted to explore the mobility influencing factors on multilane highways, it was impossible to mention all the studies due to space limitations. Twenty mobility influencing factors were identified from the literature separately for partially controlled-access highways and open-access highways. The identified factors were also rated based on their relative importance in the literature using a qualitative scale of low to high [29] (Table 2). For example, a factor was rated as "high" based on the (1) number of times it appeared in the literature and the (2) impact of the factor described in each study.

**Table 2.** Comparison of Mobility Influencing Factors for Different Highway Types.

| S/No | Factor | Impact of Factors on Highway Mobility | |
|---|---|---|---|
| | | **Partially Controlled Access** | **Open Access** |
| 1 | Access Density | HIGH | HIGH |
| 2 | Road Width | MEDIUM | MEDIUM |
| 3 | Speed limit | HIGH | LOW |
| 4 | Heavy Traffic | HIGH | HIGH |
| 5 | Roadside development | MEDIUM | HIGH |
| 6 | Median type | MEDIUM | - |
| 7 | Pavement condition | HIGH | HIGH |
| 8 | Roadside parking | MEDIUM | MEDIUM |
| 9 | Road signs | MEDIUM | LOW |
| 10 | Roadside Shoulder | MEDIUM | HIGH |
| 11 | Sidewalk | MEDIUM | LOW |
| 12 | Geometry | HIGH | MEDIUM |
| 13 | Roadside object | HIGH | HIGH |
| 14 | Driver behavior | MEDIUM | HIGH |
| 15 | Vehicle characteristics | HIGH | - |
| 16 | Lateral clearance | MEDIUM | MEDIUM |
| 17 | Pedestrian Traffic | MEDIUM | HIGH |
| 18 | Bus Stops | MEDIUM | HIGH |
| 19 | U-turns | HIGH | - |
| 20 | Slow vehicles * | - | HIGH |

**References:** [30–57], * Bicycles, Rikshaws, Motorbikes, and Non-Motorized Vehicles.

The literature review showed that most factors that impact the mobility of partially controlled-access multilane highways were similar to open-access multilane highways. However, the impact of different factors varied across the two highways (Table 2). The impact of pedestrian traffic, bus stops, driver behavior, and roadside development was high on open-access multilane highways compared to partially controlled-access highways. This difference was due to frequent pedestrian crossings, undesignated bus stops, highly heterogeneous traffic, and high access density on open-access multilane highways. Moreover, such highways mainly exist in developing countries where road user awareness and attitude toward rules and regulations differ from those in developed countries. Such distinctive characteristics differentiate open-access highways from partially controlled-access highways. Thus, it is necessary to investigate open-access multilane highways separately.

Numerous researchers have developed speed- and travel-time-based mobility models using field data on different categories of highways. Semeida [23] developed a relationship between operating speed (V85) and roadway factors and found that the most influential variables on V85 were the pavement width, the median width, and the existence of side access. Similarly, Teng and Lin [58] developed average free-flow speed models for multilane rural and suburban partially controlled-access highways. The authors identified vehicle type, speed limit, and spacing between signalized intersections as the most critical variables governing free-flow speed. Aronsson and Bang [59] also studied different factors impacting mean speed on arterial, suburban, and urban partially controlled-access highways. The authors found that factors, such as a high number of pedestrians, bicycles, and vehicle traffic flow, significantly reduced the mean speed on urban roads. Semeida and El-Shabrawy [25] explored the influence of pavement condition, roughness, and longitudinal grade on passenger car operating speed (V85) using three different modeling approaches and identified international roughness index (IRI) as the most critical parameter. Wang [60] developed a relationship between the drivers' speed and the road environment on partially controlled-access highways. The study concluded that roadside objects, including trees and utility poles, access density, driveway and intersection densities, number of lanes, lane width, on-street parking and sidewalk presence significantly influenced drivers' operating speeds.

Several observations and conclusions could be drawn from the reviewed studies and models. First, in previous research studies, various mobility influencing factors were

studied separately. However, to accurately predict travel conditions, it is important that the maximum number of mobility influencing factors be evaluated simultaneously. Secondly, past mobility-related research focused on access-controlled or partially controlled-access highways. Fewer studies have been conducted to explore the mobility of open-access multilane highways especially for developing countries. Moreover, there is a lack of a comprehensive research to determine the association between highway mobility and its influencing factors along open-access multilane highways, based on both road users' opinions and in-service road data. Therefore, this research used a multidimensional approach of analyzing both road users' opinions and field observations to explore factors impacting open-access multilane highway mobility, which is the novelty of this research. An attempt has been made to include the maximum number of significant factors in the analysis to study their combined impact on open-access multilane highway mobility.

### 3. Research Methodology

A comprehensive research methodology consisting of three major phases was adopted to achieve the objectives and bridge the identified research gaps (Figure 1).

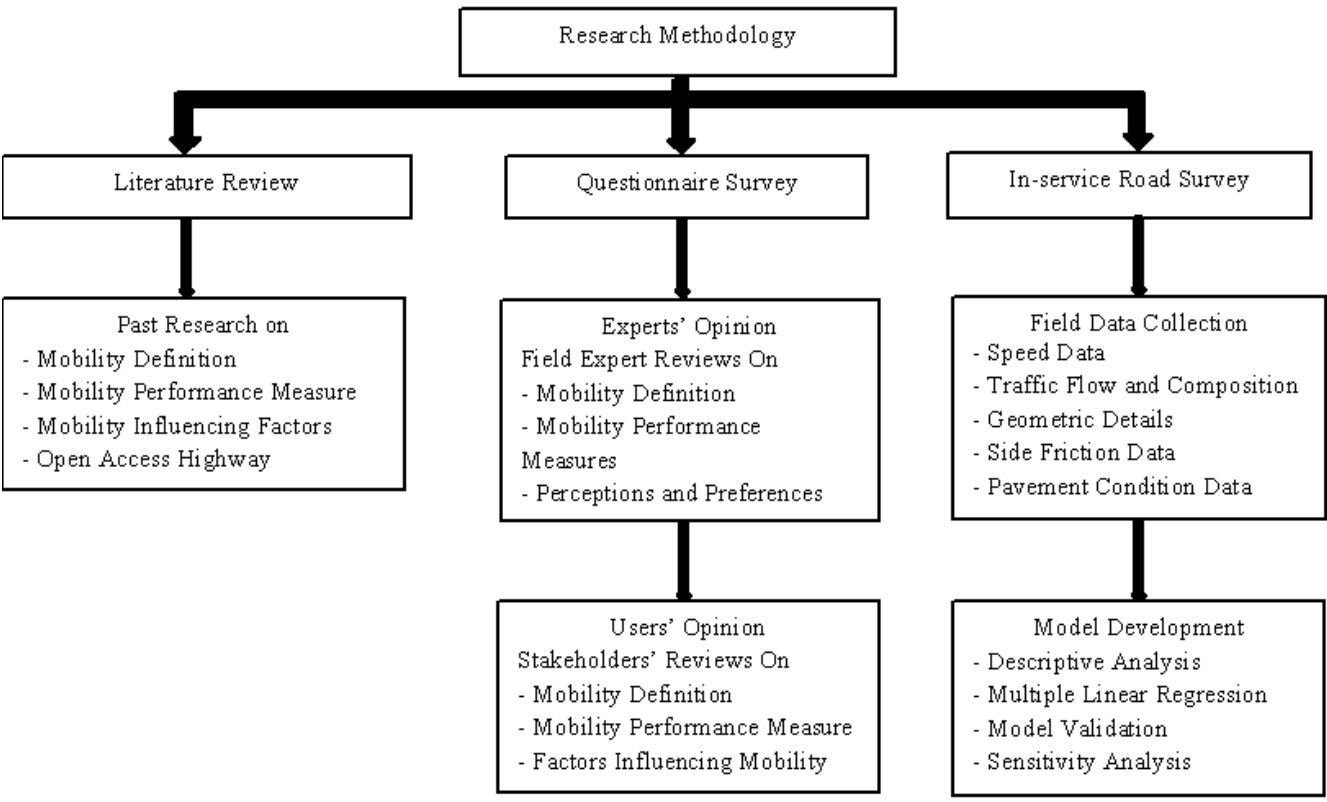

**Figure 1.** Overview of the Research Methodology.

The research started with a literature review of the current (open-access and partially controlled-access) multilane highway mobility definition, measurement, and factors influencing mobility. In the second phase, a preliminary questionnaire was developed based on the knowledge gathered from the literature review, and a pilot survey was conducted among selected field experts. In the next step, road users and people from different fields (agency officials, designers, and related educationists) were personally interviewed to answer the research questions. The data were analyzed using the statistical software SPSS to determine the most important mobility influencing factors. The one-way analysis of variance (ANOVA) test was also performed to evaluate the statistical difference of opinion among different survey groups. Based on the results of the above investigation, detailed

field data were collected from an open-access multilane highway. Finally, a statistical model was developed relating highway mobility and its influencing factors.

## 4. Questionnaire Survey

In order to explore experts' and users' perspectives on mobility, an interview-based questionnaire survey was conducted in two phases: an expert survey and a user survey. During the expert survey, a total of 24 topic experts representing different departments of transportation (10), academia (5), consultants (5), and highway management agencies (4) were interviewed. This expert survey was conducted for two reasons: (1) to explore the current definition of highway mobility for open-access multilane highways and to investigate mobility performance measures, and (2) to validate and finalize the main questionnaire (user questionnaire) for this research.

At the start of the interview, experts were presented with a list (Table 1) of mobility performance measures to select the most relevant and important ones for open-access multilane highways. Experts also revealed their views about the state of practice with regard to the use of mobility performance measures for open-access multilane highways. Finally, experts ranked selected performance measures based on predefined questions. A preliminary user questionnaire was also presented to experts for validation based on the most widely acknowledged Content Validity Index (CVI) method. The user questionnaire clearly stated that this study is particular to open-access multilane divided highways and consisted of two parts. Part I had a few qualitative questions, whereas Part II entailed ranking mobility influencing factors. Four primarily used performance measures, i.e., speed, travel time, LOS, and capacity, were selected for ranking against each factor. However, experts suggested combining interrelated performance measures, i.e., speed and travel time, and capacity and level of service. The content validity index was calculated based on relevancy scale values ranging from 0 to 1. Based on the rating by all experts, a question was considered relevant when CVI > 0.80, needs revision when the value is between 0.70 and 0.80, and eliminated if the value is below 0.70 [61]. The overall average CVI of the questionnaire was 0.87, which was above the threshold of 0.80, indicating its appropriateness. For Part-I, all questions were rated above the threshold of 0.7, so none of the questions were removed. However, for Part II of the user questionnaire, factors rated lower than the threshold (0.7) were removed. All other factors with CVI scores between 0.71 and 0.80 were analyzed to incorporate experts' recommendations. Experts also suggested adding two new factors to the user questionnaire. The first factor was "Cat-eye" (transverse raised pavement markers), which is used as a speed breaker on local highways in some urban highway sections to allow pedestrians to cross. The second factor was the "familiar drivers' population," a part of the Highway Capacity Manual (HCM) level of service methodology. Warning signs and auxiliary lanes were also added separately based on experts' recommendations. Twenty mobility influencing factors were finalized for the user questionnaire. An overview of the finalized user questionnaire is shown in Table 3. Each factor was rated against two mobility measures separately, i.e., speed/travel time and level of service/capacity. The interrelated performance measures, i.e., speed and travel time, and capacity and level of service, were combined for rating by users. A quantitative scale of 0–10 was used for this purpose, where "0" meant no impact and "10" meant a high impact on mobility. Respondents were also allowed to comment on any part of the questionnaire.

**Table 3.** Overview of User Questionnaire.

| **Multilane Highway Mobility Survey** | | |
|---|---|---|
| This questionnaire is developed to identify the factors which effect the mobility of **open access multilane divided highways (For Example N–5 GT Road)** in Pakistan. Mobility of highway in this survey refers to ability to move quickly, easily, and cheaply on a highway at a speed that represents free flow or comparably high- quality conditions. This survey has two parts, first part is related to drivers' information and understanding of highway mobility. Whereas second part is related to identification of mobility influencing factors. Please contribute to this survey using your experience. Your response to this survey is highly appreciated. | | |
| **PART–I** | | |
| **Questions** | **Options** | **Answer** |
| Name | | |
| Age | | |
| Gender | Male/Female | |
| Education | <Matric, FSC, BSC, MSC OR Higher | |
| Vehicle you drive | Car, Taxi, Van, Bus, Truck | |
| Driving Experience | <5, 5–10, 1–20, >20 **(years)** | |
| How much you know about "traffic rules, regulations and road signs"? | 100%, 75%, 50%, 25%, 0% | |
| What is "average distance you travel" on multilane divided highways? | <50, 50–100, 100–200, >200 **(km/day)** | |
| How do you define "mobility"? | Speedy travel, Free flow movement, Travel comfort, less delay | |
| What is your "criteria" of a good or bad condition of travel on GT road? (Performance measure) | Travel time, Travel speed, LOS, Capacity | |
| **PART–II** | | |
| Rate the factor in j column against performance in column i on a scale of 1 to 10. Scale of 1 Means **"Little or no impact"** whereas 10 means **"very high impact"**. | | |
| **Performance Measure (i)** | **Factor (j)** | **Rating (1 to 10)** |
| | Factor 1 | |
| | Factor 2 | |
| **Measure 1, Measure 2** | Factor 3 | |
| | Factor . . . . . . n | |

### 4.1. Survey Data Collection

In Pakistan, getting driver reviews is difficult due to the variability in driver characteristics based on their level of education. Therefore, it was decided that each participant should be personally interviewed in a roadside survey to obtain a complete and sincere response. It also ensured the inclusion of a variety of survey respondents, mainly commercial vehicle users. A total of 400 individuals with diverse sociodemographic backgrounds were interviewed. Survey respondents were divided into four groups: LTV (light transport vehicle) drivers, HTV (heavy transport vehicles) drivers, agency officials, and educationists. The category of LTV drivers included respondents of private cars, taxis, and wagons, whereas HTV drivers included respondents of buses, trucks, and trailers. The percentage of each group is shown in Figure 2. Salient details of respondents include: (a) 89% were male; (b) 63% had a driving experience between 5 and 20 years: (c) 72% of the survey respondents had adequate knowledge of traffic rules and regulations; (d) 57% of the participants reported driving more than 100 km per day on open-access multilane highways, which ensured the efficacy of collected data.

### 4.2. Reliability Test

Cronbach's alpha is considered an adequate measure of internal data consistency. A low Cronbach's alpha indicates a lack of correlation between the items on a scale. Generally, a questionnaire with an $\alpha$ value between 0.60 and 0.70 is satisfactory, whereas values greater than 0.70 are usually considered good and reliable [62]. Cronbach's alpha value for the questionnaire was 0.86, thereby confirming good reliability.

**Percentage of Survey Respondents**

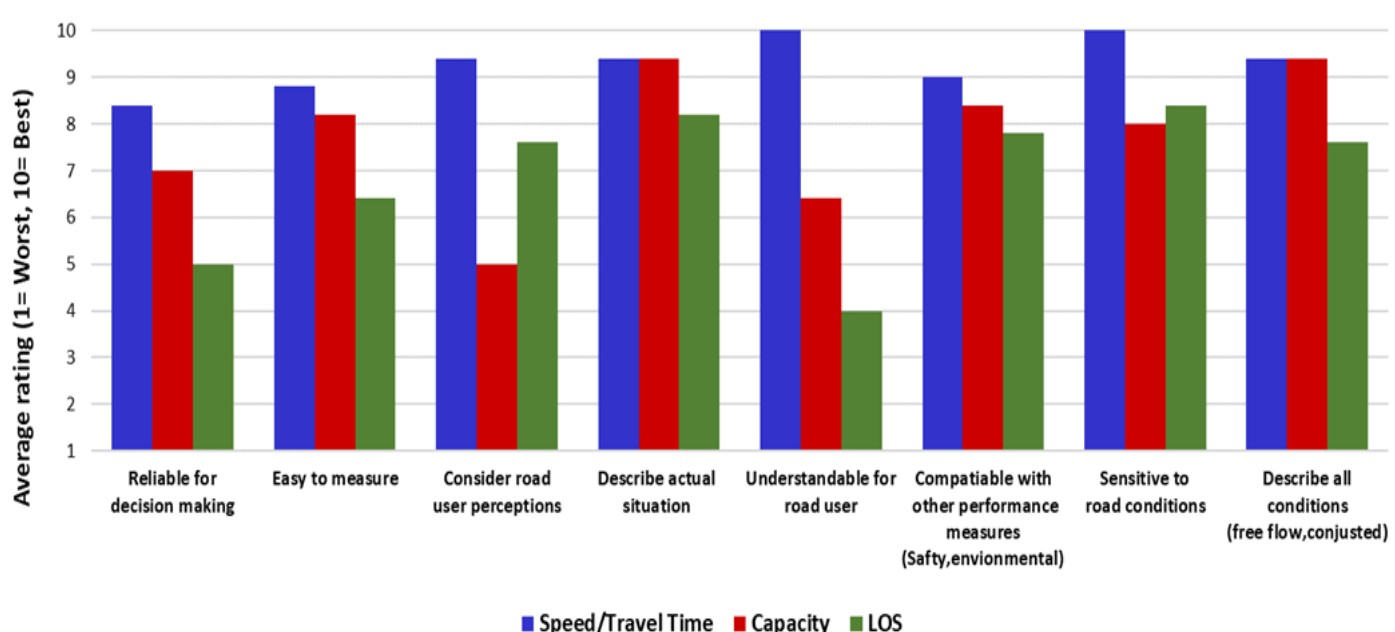

**Figure 2.** Percentage of Survey Respondents.

## 5. Questionnaire Survey Results

### 5.1. Expert Survey Results

Experts were interviewed regarding their perception of highway mobility and different mobility measures for open-access multilane highways. According to most respondents, travel time and speed are related performance measures and could be used interchangeably. Therefore, speed and travel time were taken as a single performance measure for further analysis. A mixed response was received about the definition of open-access multilane highway mobility; at the same time, some were willing to describe mobility in their own words as "movement from one location to another location as fast as possible" or "moving with a constant speed." According to experts, LOS (60%) is the most applied performance measure for open-access multilane highways, followed by speed and travel time (25%). However, the ranking of performance measures by the same experts showed that speed/travel time is more suitable than the other two measures (Figure 3). When questioned about this discrepancy, agency officials explained two reasons. First, LOS is adopted at the agency level to ensure uniformity in decision-making. Secondly, no substantial research has been conducted on other performance measures for open-access highways in local conditions that substitute the LOS methodology.

**Figure 3.** Expert Ranking of Mobility Performance Measures.

Ranking results revealed that speed/travel time is a better performance measure among the three in terms of being sensitive to road conditions, ease of measurement, and describing the actual situation on the highway (Figure 3). Moreover, speed/travel time is compatible with other decision-making criteria, such as safety analysis and economic decision-making. Experts also recognized speed/travel time as the most understandable performance measure for road users. The above discussion established speed/travel time as a stronger candidate, as a mobility performance measure of open-access multilane highways. Respondents also highlighted a few limitations of the LOS methodology, both general and specific to open-access multilane highways:

- LOS letter grades give no further information between grades, especially once LOS "F" is reached.
- The impact of pavement condition is not considered in the LOS methodology, which is a major limitation.
- Prevalent issues of open-access highways, i.e., pedestrian crossings, roadside parking, and heterogeneous traffic, are not addressed in the LOS methodology.
- The LOS methodology is difficult to understand for various stakeholders involved in policy and decision-making.

*5.2. User Survey Results*

During the second phase of survey data collection, different road users and stakeholders were interviewed based on a systematically built user questionnaire. A mixed response was received when participants were asked about mobility definition (speedy travel = 26%; less delay = 32%; travel comfort = 32%; free flow movement = 10%). However, 80% of the respondents mentioned that they were more concerned about speed or travel time than LOS or capacity while traveling on the highway. In their opinion, disaggregate measures such as speed and travel time that directly measure the traffic situation are more practical than aggregate measures such as LOS. Figure 4 shows the rating analysis of different factors on speed and LOS, separately and collectively. The overall mean rating value for all the studied mobility influencing factors was 6.53 when rated against speed/travel time, whereas this value was 6.47 for LOS/capacity.

Average rating results revealed that heavy traffic is the most critical mobility influencing factor, followed by road width and condition. Moreover, the top rank of "heavy vehicles" indicated that even if road width is increased, heavy vehicles will still be more influential on the mobility of open-access multilane highways. The "Slow vehicles" factor was also highly rated along with "heavy vehicles," though the latter's impact is greater than the former. The survey respondents reported that pedestrians usually cross the highway at undesignated places on open-access multilane highways. Therefore, respondents rated pedestrian crossings among the top ten mobility influencing factors. Respondents mentioned that excessive U-turns by two-wheelers and three-wheelers at undesignated places directly impact highway mobility. Therefore, "U-turn" was also highly rated by the survey respondents. Rating analysis revealed that the essential LOS-related factor, "familiar drivers' population," was not among the top ten influential factors. In contrast, "pedestrian traffic" was considered more important on open-access multilane highways, which is not a part of the LOS methodology. From the above analysis, it was concluded that the mobility of open-access multilane highways mainly depends on roadside resistance (access density, pedestrian crossing, and roadside parking), traffic composition (heavy vehicle and slow vehicles), and road layout (road width, pavement condition, and U-turns).

This study also explored the association of different performance measures with users' perceptions. It was found that factors directly impacting instantaneous speed (heavy traffic, slow vehicles, pedestrian traffic, and road condition) were highly rated by drivers when considering speed/travel time as a performance measure (Figure 4). However, for LOS/capacity, all those factors that reduce highway capacity (lane width, roadside parking, and roadside shoulder) were highly rated. A one-way ANOVA test was also performed on data to compare the means of different groups (using the F-distribution). The results

revealed a significant difference of opinion ($p < 0.05$) on pavement condition, slow vehicles, pedestrian traffic, cat eyes, geometry, median type, and roadside objects when rated against two performance measures (Figure 4) [41]. The above discussion established that the importance of mobility impacting factors also depends on the performance measures used for analysis.

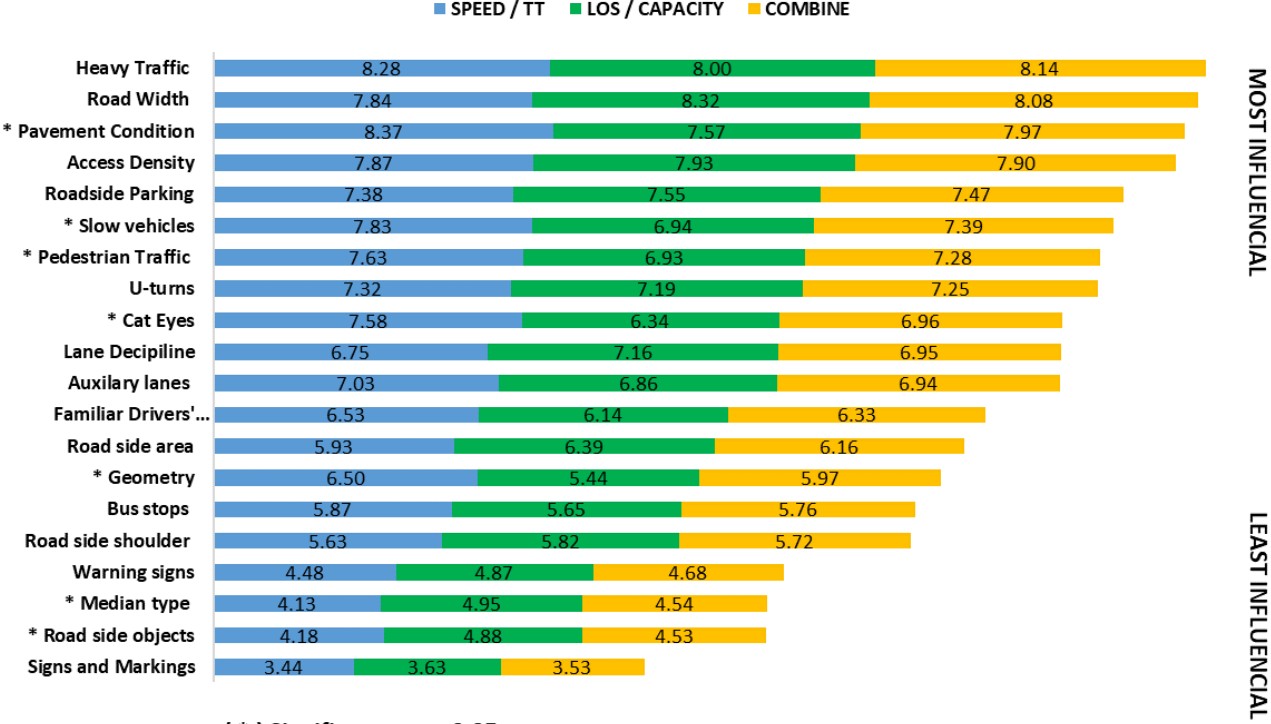

**( * ) Significant at $p < 0.05$**

**Figure 4.** Ratings of Mobility Influencing Factors (Scale 1–10).

### 5.3. Ranking Analysis of Survey Groups

As mentioned earlier, survey respondents were divided into four groups to explore their opinion regarding mobility influencing factors. Comparative analysis of survey groups revealed the following significant differences (Figure 5).

- A comparison of rating results of HTV and LTV drivers showed that most design-related factors (road width, pavement condition, median type, and geometry) were highly rated by the former than the latter. In contrast, roadside friction factors (pedestrian traffic, bus stops, and roadside area) were highly rated by LTV drivers compared to HTV drivers.

- Agency officials and designers also had distinct opinions regarding a few mobility influencing factors. Agency officials reported a high association of highway mobility with roadway design factors (road width, median type, and U-turns), traffic composition (heavy vehicles and slow vehicles), and pedestrian traffic. In comparison, designers and educationists considered pavement condition and roadside development more influential on highway mobility.

- Another comparison of agency officials with LTV and HTV drivers also revealed a difference of opinion. "Roadside parking" and "shoulder" were considered more influential (mean rating 7.9) by LTV and HTV drivers as compared to what agency officials have cited (mean rating 6.25). Similarly, LTV and HTV drivers rated roadside objects as least influential (mean rating 4.0), contrary to what agency officials have mentioned (mean rating 6.25).

- Further analysis was carried out using one-way ANOVA tests to determine the significant difference of opinion ($p < 0.05$) between different survey groups. Results revealed that HTV and LTV drivers disagreed ($p < 0.05$) on fourteen factors (mentioned with * in Figure 5). Similarly, a significant difference of opinion ($p < 0.05$) between agency officials and designers was also observed on pavement condition, pedestrian traffic, bus stops, signs and markings, and geometry (mentioned with + in Figure 5) [41].

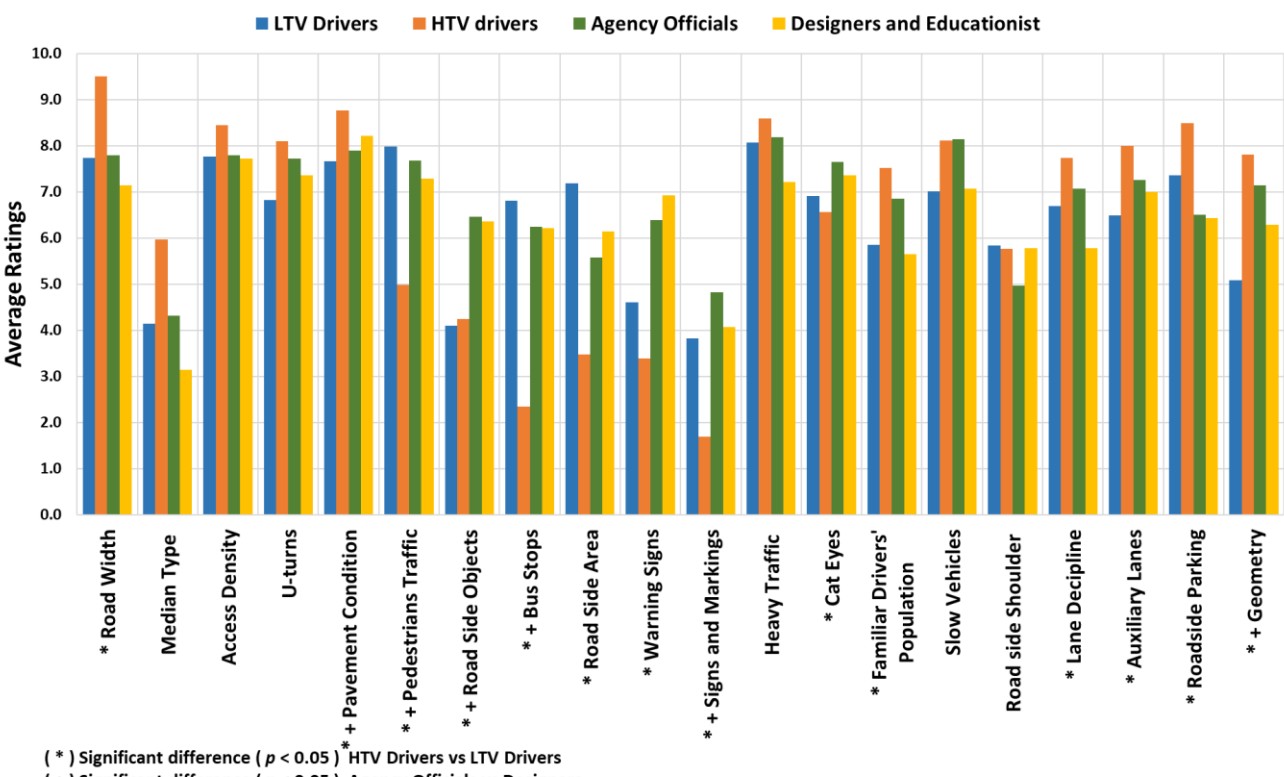

**Figure 5.** Difference of Opinion Between Survey Groups.

## 6. In-Service Road Survey

According to the scope of this study, National Highway (N-5) was identified as a typical example of an open-access multilane highway. A corridor of 850 km was selected, forming a geographically representative sample of the country's open-access multilane highway network. For this study, only the level terrain of highway sections was selected, and all the non-tangent locations were removed from the total length of the study area. Moreover, to avoid the impact of the speed limit on the traffic flow, only those sections were selected where the speed limit remained constant (i.e., 100 km/h). A highway section/site was defined as a stretch of the road having a length of 500 m. The following site selection criteria were used.

- Each section should have an open-access environment.
- Each section should be free from the impact of horizontal and vertical grades.
- Median type and the number of lanes should not vary within a section.
- Areas near grade intersections, prone to congestion, should be avoided.

The dataset assessed in this study was collected by the National Highway Authority (NHA) on national highway N-5. The selected highway is a true representative of an open-access multilane highway network of Pakistan. The base data included a video photographic survey that contained georeferenced images of the highway using a survey system attached to a test vehicle. These video recordings also displayed the instantaneous

speed of the test vehicle after each 100 m segment of road, using a GPS device attached with the survey system. For each section of 500 m, the average speed of the test vehicle was recorded manually by observing the video, which was taken as representative speed of the traffic stream. Multiple drivers drove the test vehicle on different days of the week, reducing the bias of the collected data. The additional data attributes, i.e., pavement condition and road width, were also gathered from the National Highway Authority database. For each section, speed data and associated road features were recorded in a Microsoft Excel workbook. This database was then used in analysis and model development.

### 6.1. Statistical Model Development

As discussed in previous sections, average travel speed was identified as the most suitable performance measure and was used as the response variable in the following regression analysis. Independent variables in the model included roadside friction elements, cross-section features, and on-road features. Roadside friction contained elements of the built environment and activities directly adjacent to the highway, i.e., pedestrian activity, access density, and roadside parking. In this model, roadside friction also served as a proxy for roadside development. Cross-sectional features included the number of lanes, median type, service road, pavement condition, and U-turn. As the pavement condition data were acquired from the National Highway Authority (NHA), international roughness index (IRI) was used an indicator of pavement condition. IRI is a well-established and globally used/accepted indicator of pavement condition/ride quality and is likely to capture the effects of major pavement distresses such as cracking, spalling, faulting, and rutting [63–67]. Most of the selected highway sections for this study were characterized by unpaved shoulders, undesignated walkways, access ramps, and parking lanes. Therefore, these aforementioned factors were less significant in the case of this study. On-road traffic features included traffic volume, density, and vehicle fleet composition. The vehicle fleet comprised passenger cars, heavy trucks, and slow-moving vehicles. Table 4 presents the list of investigated independent variables.

**Table 4.** Description of Independent Variables.

| S/No | Variables | Description | Min | Max | Average | SD |
|------|-----------|-------------|-----|-----|---------|-----|
| 1 | Flow | Vehicles/hour | 120 | 4355 | 1528 | 843 |
| 2 | Heavy Vehicles | Vehicles/hour (% of total flow) | 0 | 59.7 | 10.6 | 12.9 |
| 7 | Slow Vehicles | Vehicles/hour (% of total flow) | 0 | 20 | 2.39 | 2.74 |
| 3 | Access Points | Number of access points/500 m | 0 | 12 | 1.50 | 1.59 |
| 8 | Pedestrian along | Number of pedestrians/500 m | 0 | 28 | 1.30 | 2.82 |
| 4 | Pedestrian crossings | Number of pedestrians/500 m | 0 | 27 | 1 | 2.23 |
| 5 | Parking | Number of parked vehicles/500 m | 0 | 37 | 0.5 | 2.15 |
| 6 | Pavement condition | International roughness index (IRI) (m/km) | 0.73 | 5.57 | 2.70 | 1.25 |
| 9 | Median Type | Green belt, Jersey barrier | GB = 71.5% | | JB = 28.5% | |
| 10 | Number of lanes | 4 Lanes, 6 Lanes | 4 lanes = 83.5% | | 6 lanes = 16.5% | |
| 11 | U-turn with turning lane | Present, Absent | Present = 26.2% | | Absent = 73.8% | |
| 12 | U-turn without turning lane | Present, Absent | Present = 10.8% | | Absent = 89.2% | |
| 13 | Service Road | Present, Absent | Present = 12.5% | | Absent = 87.5% | |

Initially, scatter plots were used to check the correlation between dependent variables and each of the independent variables. Two of the measured variables were categorical, which included median type and number of lanes. For median type, a value of 1 is used if the median is green belt and 2 if it is jersey barrier. Data also included binary variables that indicate the presence or absence of a roadway feature. For example, for the service road, a binary variable was used that takes the value of 1 if the service road is present and 0 if it is not.

The independent plots showed a linear relationship between the majority of dependent and independent variables. In addition, nonlinear regression, with numerous iterations, was attempted but significant variables and overall model statistics were found to be less satisfactory compared to the multilinear regression model form. Therefore, the multiple

linear regression model was considered for use to best describe the relationship between dependent and independent variables. The data were modeled using a statistical software package for the social sciences (SPSS). The model followed a stepwise procedure, and the iterative process continued until all variables had a significance of 95% or higher. Finally, the model with the smallest number of independent variables, minimum RMSE (Root-Mean-Square Error), and highest $R^2$ (coefficient of determination) value was selected [31].

### 6.2. Model Results

Multiple linear regression analysis of the observed average speed was performed, and the resulting best model for an open-access multilane highway was as follows:

$$SPEED = 86.655 - 4.014 * AD - 0.007 * FL - 1.354 * PC - 1.112 * MT - 0.185 * HV - 0.828 * IRI + 5.518 * NOL \quad (1)$$

where:

*AD*: Access density (Access points/500 m);
*FL*: Flow (Vh/h);
*PC*: Pedestrian crossing (Pedestrian/500 m);
*MT*: Median type (Green Belt, Jersey Barrier);
*IRI*: International Roughness Index (m/km);
*HV*: Heavy vehicles (% of total flow);
*NOL*: Number of lanes (Four, Six).

The model yielded an $R^2 = 0.74$ (Adjusted $R^2 = 0.73$, RMSE = 6.72), and all the independent variables included in the model were observed to be statistically significant at the 0.01 level ($p < 0.01$). Insignificant variables were U-turns with turning lanes, U-turns without turning lanes, service roads, and roadside parking. The signs of the multiple linear regression coefficients showed that all independent variables are negatively correlated with speed except the "number of lanes." Intuitively, an increased number of lanes would lead to greater travel speeds. Finally, multicollinearity was investigated by the most widely used variance inflation factor (VIF). Multicollinearity is a common problem in linear model development. High correlations among predictor variables lead to unreliable and unstable estimates of regression coefficients. When VIF is equal to 1, there is no multicollinearity among regressors. An VIF between 1 and 5 shows variables that are moderately correlated, and an VIF greater than 5 means variables are highly correlated. Generally, it would be a concern when the VIF value for a predictor is greater than 2.5 [68,69]. The results illustrated in Table 5 show that the VIF values were lower than 2.5, which confirmed that there was no serious multicollinearity among the predictor variables.

**Table 5.** Summary of Regression Results.

| Factor | Coefficients | Std. Error | T-Stat | Sig. | Collinearity (VIF) |
|:------:|:------------:|:----------:|:------:|:----:|:------------------:|
| AD | −4.014 | 0.141 | −28.480 | 0.000 | 1.234 |
| FL | −0.007 | 0.000 | −26.260 | 0.000 | 1.342 |
| PC | −1.354 | 0.092 | −14.772 | 0.000 | 1.111 |
| HV | −0.185 | 0.016 | −11.548 | 0.000 | 1.035 |
| NOL | 5.518 | 0.700 | 7.885 | 0.000 | 1.632 |
| IRI | −0.828 | 0.168 | −4.915 | 0.000 | 1.093 |
| MT | −1.112 | 0.565 | −1.969 | 0.049 | 1.440 |

### 6.3. Model Validation

Before model development, the data were randomly segmented into two datasets: one for calibrating the model (80%) and the other for validating the developed model (20%) [70]. The validation was performed by comparing the predicted speed values computed using the developed model with actual speed values observed on the field. The comparison result indicated that the predicted speed values agreed with the observed values (Figure 6). The R-square value of 0.75 between the observed and predicted values indicated that the model

is valid [69]. The predictive capability of the developed model was also tested by calculating the mean squared prediction error (MSPE) using the developed model parameters and the new dataset by comparing it with the mean squared error (MSE) of the developed model. The model MSPE (8.77) was close to the MSE (7.66), demonstrating that it yields satisfactory validation results [69].

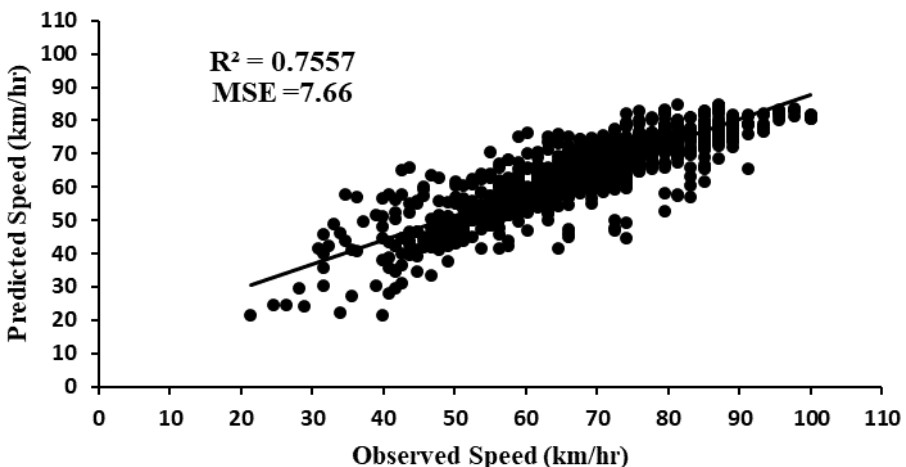

**Figure 6.** Observed vs. Predicted Plot for Developed Model.

### 6.4. Model Application and Sensitivity of Variables

The developed model can be particularly used in road design and rehabilitation, to improve highway mobility. Additionally, it can provide speed predictions to assist long-term highway management strategies without spending a heavy budget on traffic monitoring equipment. The beta coefficients (β) of the independent variables in Equation (1) showed that with an increase in one access point per 500 m length of the highway, travel speed is expected to be reduced by 4 km/h. Similarly, with every increase in crossing pedestrians, travel speed is expected to be reduced by 1.4 km/h. Therefore, at first, highway mobility in terms of travel speed could be improved by using economical solutions, i.e., controlling access points and pedestrian crossings. However, at a later stage, mobility could be improved by adding more lanes. The beta coefficient (β) of "Number of lanes" in Equation (1) confirmed that travel speed increased by 5.5 km/h as the number of lanes increased from four to six lanes.

The developed model was also tested for practical application by varying the value of any one selected independent variable, while keeping the values of all other independent variables constant at reasonably good values, as observed from the field data. For reasonable expected values (FL = 2000, IRI = 2.5, HV = 10, and PC = 2) the mobility of the highway in terms of speed could be improved from 46 km/h to 62 km/h by reducing the access points from 5 to 1 per 500 m, on a four-lane divided highway. Similarly, the speed could be increased from 56 km/h to 60 km/h by improving the IRI value from 4 m/km to 2 m/km. In this manner, the developed model could be utilized by highway agencies to address the mobility and accessibility issues, both for newly constructed as well as in-service open-access multilane highways.

The sensitivity of each explanatory variable was also explored based on standardized regression coefficients in the finalized model (Figure 7) [23]. Analysis showed that the most significant independent variable impacting the speed was access density (0.48), followed by the flow (0.46) and then pedestrian crossing (0.23). In contrast, median type was the least significant variable. Though the significant factors identified in this model conformed to the literature findings, the sensitivity of factors still revealed valuable information. "Access density" had the highest impact on open-access multilane highway mobility compared to any other factor. Moreover, pedestrian crossing had a greater influence on mobility than

other important factors, i.e., the number of lanes and IRI. Comparison of the sensitivity results with road users' opinions also revealed an inconsistency. According to road users, the heavy vehicle was the most significant variable, followed by road width and pavement condition (IRI) (Figure 4). However, modeling results revealed that these factors had a lesser impact on travel speed than access density and pedestrian crossing (Figure 7). Similarly, "median type" was rated as the least important factor by survey respondents, opposing actual field observation.

**Figure 7.** Sensitivity of Explanatory Variables.

## 7. Conclusions and Recommendations

Open-access multilane highways differ from access-controlled and partially controlled-access highways, especially in developing countries. This study provided a synthesis of open-access multilane highway mobility and its influencing factors based on road users' perceptions as well as the in-service road data, which was the novelty of this research. The following paragraphs summarize findings based on the literature review, survey results, and regression analysis.

- Most mobility influencing factors were common across partially controlled-access and open-access multilane highways. However, the impact of different factors varied across the two highways. In addition, mobility influencing factors are multiple, which need to be evaluated simultaneously to accurately predict travel conditions.
- An expert survey revealed that no substantial research has been conducted on mobility performance measures for open-access highways in local conditions. Moreover, experts mentioned that the applicability of the level of service methodology to find the performance of open-access multilane highways in developing countries has the following issues.

  - LOS letter grades give no further information between grades.
  - The impact of pavement condition is not considered in the LOS methodology.
  - Prevalent issues of open-access multilane highways, i.e., pedestrian crossings, roadside parking, and heterogeneous traffic, are not addressed in the LOS methodology.

- Mobility on open-access multilane highways could be explained as "continuous and speedy travel with less delay." Moreover, based on experts' and road users' opinions, speed and travel time are the most suitable performance measures for analysis and decision-making on open-access multilane highways. This answered the first research question of the study.
- Results of the questionnaire survey and in-service road survey revealed that high access density, frequent pedestrian crossings, and traffic heterogeneity have greatly reduced mobility on open-access multilane highways, in addition to road width and

pavement condition. This confirmed that open-access multilane highways have a more complex and unique driving environment than partially controlled-access highways.

- Questionnaire survey results revealed that an essential LOS-related factor, "familiar drivers' population," was not amongst the most significant factors impacting the mobility of open-access multilane highways. In comparison, "pedestrian traffic" and "slow vehicles" factors were more significant, which are not present in the LOS methodology.

- A significant difference of opinion was found between service providers (Agency Officials and Designers) and road users (HTV and LTV drivers) on important factors, including pavement condition, pedestrian traffic, and highway geometry. Moreover, it was also revealed that the importance of mobility influencing factors depends upon the performance measure used for analysis. Therefore, it is suggested that all the stakeholders' opinions regarding the mobility performance measure and mobility influencing factor should be considered in developing policies and the highway planning process.

- A comparison of rating analysis with model results revealed that the road users' opinions about the significance of mobility influencing factors were inconsistent with the actual field observations. However, it was established that the mobility of open-access multilane highways is primarily influenced by high access density, frequent pedestrian crossings, unsatisfactory pavement condition, and traffic heterogeneity.

The last four conclusions answered the second and third research questions raised at the start of this study. These findings of this study would surely contribute to the existing body of knowledge and provide valuable information to highway agencies, transportation planners, and designers for designing and managing open-access highways. Moreover, this study would also help to improve highway design in Pakistan as the developed model included additional mobility-influencing factors compared to previously available models. For future research, it is recommended that the additional factors identified in this research should be incorporated into the level of service (LOS) methodology that would improve its effectiveness and applicability for open-access multilane highways. Furthermore, a procedure should be developed to integrate road users' feedback in highway development and improvement projects.

**Author Contributions:** Conceptualization, J.A.K., M.B.K. and A.H.; methodology, J.A.K., M.B.K., A.H. and A.A.; software, J.A.K. and M.B.K.; validation, J.A.K. and M.B.K.; formal analysis, J.A.K., M.B.K. and A.H.; investigation, J.A.K., M.B.K. and A.H.; resources, J.A.K., M.B.K. and A.H.; data curation, J.A.K., M.B.K. and A.A.; writing—original draft preparation, J.A.K. and M.K; writing—review and editing, J.A.K. and M.B.K.; visualization, J.A.K., M.B.K. and A.H.; supervision, J.A.K., M.B.K., A.H. and A.A.; funding acquisition, J.A.K., M.B.K., A.H. and A.A. All authors have read and agreed to the published version of the manuscript.

**Funding:** This research received no external funding.

**Data Availability Statement:** Not applicable.

**Conflicts of Interest:** The authors declare no conflict of interest.

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
