# Peer review of "A Multidimensional Analysis of Factors Impacting Mobility of Open-Access Multilane Highways"

_infrastructures, doi:10.3390/infrastructures7100143_

Round 1

Reviewer 1 Report (New Reviewer)

The topic is exciting but the real contribution needs to be clearly presented. Moreover, in the model results section (6.2), the outcome of the SPSS should be shown including the different models with their indicators such as R2!

I would like to see the three research questions stated clearly not like what you have in lines (67-70). It should be linked to the findings at the end.

Author Response

Reviewer 2 Report (New Reviewer)

The paper was well-written and organized. I have some minor comments on the paper:

  1. Some acronyms need to be defined/spelled out (IRI, HCM) at their first occurrence.
  2. Keep the consistency in variable names in Equation 1, Table 4, and text afterwards.
  3. What’s the definition of slow vehicles?
  4. In Figure 7, based on the context, I think it should be MSE instead of RSME? Please double check.
  5. What is the representativeness in the field data used for the multivariate regression modeling? And how was the speed data aggregated in the study, like hourly-average or 15-min average? Please elaborate more on the field data.
  6. What is the process of obtaining traffic speed from the video recordings. Any image processing methods/software used?
  7. As travel speed is limited by posted speed, my question here is whether the posted speed limit vary greatly on the study corridor. If so, this might exert significant influence on traffic flow and should be discussed in paper.

Author Response

Reviewer 3 Report (New Reviewer)

Round 2

Reviewer 1 Report (New Reviewer)

Dear Writers, I liked the changes that were made to the document. In such studies, it is critical to emphasize the importance of approaching research questions from the viewpoints of developing countries. I would like to see additional literature support for this critical issue. Here are some references for your consideration if you agree:

Potential barriers for asset management systems: a comparison between Libay and Spain

Paving the road for sustainable construction in developing countries: a study of the Jordanian construction industry 

Strategies to enhance implementation of infrastructure asset management in developing countries

Challenges and barriers to establishing infrastructure asset management: a comparative study between Libya and USA

Author Response

This manuscript is a resubmission of an earlier submission. The following is a list of the peer review reports and author responses from that submission.

Round 1

Reviewer 1 Report

Dear Authors

The topic presented is interesting but the analysis is very simple and no fruitful result is presented. There are lot of methodologies available to play with the choice dataset. These simple ratings are difficult to adopt as such. I suggest you to go through with more published literature and improve your analysis.

Reviewer 2 Report

Dear

The article deals with the issue of highway design, for which a survey was conducted based on interviews, which was to show the benefits of highways. The survey should serve to improve the design of highways in Pakistan.

• It would be useful to explain the concept of "open access highway", as motorway-type roads do not allow pedestrian access and there are no pedestrian sidewalks. This is mainly due to the safety and high speeds on these roads

• it would be appropriate to define LoS as perceived on these motorway-type roads mentioned in the article. Required elements such as speed, travel time or traffic density are common parameters that can very easily be measured today by very accurate detection, which is not mentioned.

• LoS is listed separately, but table 1 talks about other parameters, the text on pages 3 and 4 thus becomes incomprehensible

• Table 2 is incomprehensible and very general when comparing types of motorways. Nowhere is it defined what is meant by "low, medium, high". The individual types of motorways should also be clearly defined, which are also incomprehensible to readers, as most different motorways are precisely defined by parameters or even motorways are called a single road, which is set by a national and international standard. an example is that there are no bus stops on the motorways and no pedestrian movement is possible. Otherwise, such a road is not called a motorway, but a certain class of road is defined

• The survey was conducted on 24 experts who understand the issue, but no other objectifying data were taken into account.

• in Figure 3, the percentages of a "light or heavy" vehicle are used. it is not clear what the vehicles were, it is a very vague and inconclusive designation.

• The values ​​in Figure 4 should also be substantiated by exact measurements using detectors or data calculations.

• Figure 5 does not have a defined clear scale with units and there is no clear calculation for comparing the defined data if a scale of 1-10 was given.

• Some of the outputs shown in Figure 6 are obvious and more than expected on motorway roads. therefore, the benefits of the survey are not clear. many outputs can be seen in the standards of neighboring countries in Asia or on the recommendations of international institutions such as PIARC, etc.

• The conclusion is very vague and in many cases quite clearly expected among professionals on highways.

The article should be fundamentally reworked and clarified. The conclusions presented are known for many years and are also recommended by international institutions headed by PIARC. If we talk about highways, then some of the conclusions cannot be considered at all with regard to speed, safety, etc. An example is turning on the highway, walking pedestrians, bus stops, etc. The article must be clarified with regard to highway speeds, defined terms survey of vehicle weight, etc.

Best regards

Round 2

Reviewer 1 Report

Paper is acceptable in current format. Minor english editing shall be helpful.

Reviewer 2 Report

Dear

I recommend inserting an explanation of the use of motorways in the text of the article so that it is clear to the reader which motorways are being compared and why. I recommend expanding more and adding a new picture of an open highway and a standard highway for illustration. lines 34-36, 43 - 50, need to be better expanded.

1) It is important to explain why there is talk of a motorway if the vehicles are traveling at a speed of 40 km/h and in principle it is a normal service road. This point has not been incorporated

2) if the article refers to some literature, then the description and character should correspond to it, because it is not, it is necessary to define what is meant by LoS. This point has not been incorporated

3) the survey did not use any objective data from the measurement, only the form of a questionnaire. It should be justified why objective data from traffic flow measurements were not used either.

4) The division of low, meddium and high should also be specified. everything is set to 33.33% probably those levels should be defined differently see. line 170 - 172

5) if the scale 1-10 is given in the text and in the figure only 1-5, it should be consistent in the article. This point has not been incorporated

6) some of the outputs shown in Figure 6 are obvious and more than expected on motorway roads. therefore, the benefits of the survey are not clear. many outputs can also be viewed in the standards of neighboring countries in Asia or on the recommendations of international institutions such as PIARC, etc. An explanation should be given in the text. In addition, the benefits of whether to implement open motorways or not are not clear. this point has not been incorporated,

7) Conclusions and recommendations can be significantly improved. The article could be improved and refined overall

Best regards